# Informed Consent: Legal Obligation or Cornerstone of the Care Relationship?

**DOI:** 10.3390/ijerph20032118

**Published:** 2023-01-24

**Authors:** Margherita Pallocci, Michele Treglia, Pierluigi Passalacqua, Roberta Tittarelli, Claudia Zanovello, Lucilla De Luca, Valentina Caparrelli, Vincenzo De Luna, Alberto Michele Cisterna, Giuseppe Quintavalle, Luigi Tonino Marsella

**Affiliations:** 1Department of Biomedicine and Prevention, University of Rome “Tor Vergata”, 00133 Rome, Italy; 2Department of Public Health and Infectious Diseases, Sapienza University of Rome, 00185 Rome, Italy; 3Department of Clinical Science and Translational Medicine, Section of Orthopedics and Traumatology, University of Rome “Tor Vergata”, 00133 Rome, Italy; 4XIII Section, Civil Court of Rome, 00192 Rome, Italy; 5Fondazione Policlinico “Tor Vergata”, 00133 Rome, Italy

**Keywords:** informed consent, malpractice, medical errors, liability, legal, physician–patient relations, communication

## Abstract

The topic of informed consent has become increasingly important in recent decades, both in the ethical-deontological field and as a duty of law. The review covered all sentences issued by the 13th section of the Civil Court of Rome during the period January 2016–December 2020. During this period, 156 judgments were found in which a breach of consent was required; in 24 of these, specific liability was proven, and the corresponding compensation liquidated. Moreover, 80% of the cases concerned the lack of information provided. The most involved branches were those related to surgical areas: general surgery, plastic surgery and aesthetic medicine and orthopaedics. The total amount of compensation paid was EUR 287,144.59. The research carried out has highlighted how, in a broad jurisprudential context, the damage caused by the violation of the right related to informed consent is considered, and how it impacts on the economic compensation of damages. Additionally, it showed that the areas most affected by the information deficit are those related to the performance of surgical activities, which are characterized by greater invasiveness and a higher risk of adverse events. The data reported underline the exigency to consider informed consent not as a mere documentary allegation but as an essential moment in the construction of a valid therapeutic alliance, which is also useful for avoiding unnecessary litigation that is becoming increasingly burdensome for healthcare systems all over the world.

## 1. Introduction

The informed consent doctrine, with particular reference to the field of healthcare, relies on professional ethics-related aspects rather than on purely medical procedures. It is only in the second half of the last century that the informed consent doctrine became part of various legal systems, having its conceptual origins in the United States at the beginning of the twentieth century. Previously, patient clinical information was mostly left unspoken of, in accordance with the Hippocratic oath stating that physicians have to reveal nothing to their patients about their future or present clinical condition [1]. This attitude relied on a paternalistic doctor–patient relationship until a few decades ago. Over time, along with the evolution of medical science, a step forward was made in approaching the physician–patient relationship, leading to the consideration of patients as real decision-makers about their own healthcare and treatments.

One of the first legal cases concerning the acknowledgment of the patient’s right to be informed and to self-determination about his/her health dealt with the American case Shloendorff v. New York Hospital in 1914. The landmark judgment about the aforementioned judicial matter stated that “… Every human being of adult years and sound mind has a right to determine what shall be done with his own body; and a surgeon who performs an operation without his patient’s consent, commits an assault, for which he is liable in damages *…*” [2].

The first time “informed consent” appeared in a medical-related judgment was on October 22, 1957 in a malpractice judgment from California Court of Appeals—Salgo v. Leland Stanford Jr. University Board of Trustess. In this case, the court concluded that: “A physician violates his duty to his patient and subjects himself to liability if he withholds any facts which are necessary to form the basis of an intelligent consent by the patient to the proposed treatment … in discussing the element of risk a certain amount of discretion must be employed consistent with the full disclosure of facts necessary to an informed consent …” [3].

At the international level, the transposition of this principle can be related to what emerged from the famous Doctors’ Trial held in Nuremberg in 1947 against Nazi doctors, which stated the mandatory consent by the concerned party to undergo health treatments and scientific research protocols [4]. In Europe, the recognition of the above-mentioned principles may also be found in various documents, most notably the Charter of Fundamental Rights of the European Union providing that: “in the medical field and biology, the following requirements are needed: the free and informed consent by the person concerned, in the manner laid down by law”.

Respect for everyone’s right to self-determination in health care decisions has also involved some specific aspects such as the ability to make choices, the right to obtain a free, prior and informed consent of the party concerned, and the possibility to donate one’s body to medical science after death, a choice that is currently protected by specific laws in much of the world [5].

At a global level, as a result of this conceptual progress, various countries have made efforts to promote, through specific regulations, compliance both with autonomy and, consequently, with informed consent in the health field.

Looking at the Italian scenario, before the introduction of a specific law, the Italian Constitutional Court approached the informed consent issue in judgment no. 438 of 23.12.2008, providing an informed consent intended as the expression of conscious compliance with any medical treatment proposed by healthcare professionals and as a real individual’s right according to the rules stated in Art. 2 of the Constitution, protecting and promoting their fundamental rights, and in Articles no. 13 and 32 of the Italian Constitution.

Following the aforementioned judgment, Law No. 219/2017 entitled “Provisions for informed consent and advance treatment directives” was introduced in Italy, published in the Italian Official Gazette in January 2018. With this regulatory action, particular attention has been given to the doctor–patient relationship and the resulting care relationship as prerequisites to informed consent which, in turn, brings into focus both patient and doctor autonomy in clinical decision making as well as the physician’s ensuing liability [6,7]. This law also introduced into the Italian legislative framework for the first time the opportunity for each citizen, in the event of his or her future inability to self-determine, to define in advance to which treatments he or she will be subjected [8].

Therefore, it is evident that in the healthcare field, informed consent has long become an act both of legal and ethical-deontological relevance. In recent literature, the topic of the legal aspects related to informed consent has been addressed by considering not only its reflections in daily practice, but in relation to specific situations of current relevance such as the practice of telemedicine [9] and the digitization of health systems [10], biomedical research [11], and palliative and end-of-life care [12].

Regarding the duty to inform, it should be noted that information standards vary according to the legal system. For instance, two distinct models called ‘reasonable medical practitioner’ and ‘reasonable patient standard’ have been proposed in recent years. According to the first model, the amount of information should conform to what a reasonable doctor in that situation would provide, whereas in the case of the second model, the level of information should concern what a standard patient would want to know about his or her specific situation [13].

Therefore, with regard to the legal implications, the omission or lack of prior information to be provided to the patient and the obtaining of valid informed consent can be considered elements of liability for healthcare professionals and may lead to compensation for damage with the risk of being charged with damages resulting both from harm and the breach of the right to self-determination. Law no. 219/2017, Art. 1 paragraph 1 states that “… no health treatment can be started or continued without the free and informed consent of the person concerned, except in cases expressly provided for by law…” [14], and Art. 1 paragraph 4 refers to the ways by which consent must be acquired (“documented in writing or through video recordings or, for the disabled person, through devices that allow them to communicate…”), ruling the terms and procedures guidelines on informed consent. In addition, the aforementioned Law states that “the time of communication between doctors and patients constitutes healthcare time”, thus emphasizing the importance of promoting and enhancing a trusty physician–patient relationship. Moreover, according to the intention of the Italian legislator, the time spent communicating with the patient is an integral part of the health treatment path as it is always in the best interest of the patient in order to protect him/her from harm [15].

The aim of our research is to present the data relating to the phenomenon of medical malpractice resulting from non-compliance with the informed consent regulations through the analysis of the judgments from the Civil Court of Rome.

## 2. Materials and Methods

The retrospective review covered all judgments issued by the Judges of the Civil Court of Rome, XIII Section, published between January 2016 and December 2020. Only first instance judgements were taken into consideration, excluding both second instances, as well as those issued by the Supreme Court of Cassation. The XIII Section of the Civil Court of Rome deals with professional liability trials, including the medical field. The University of Rome “Tor Vergata” and the Civil Court of Rome signed an agreement, for which the court provided the judgments for analysis. The research was initially performed using the keywords “medical liability” and “medical professional”. The documents were saved in PDF format and anonymized to preserve the litigants’ personal identities and remove any connection between the tort in question and specific individuals or institutions. At the end of the anonymization phase, out of 1190 total documents (of which 23 duplicates were deleted), only 1167 underwent a preliminary analysis, performed by three different auditors skilled in medical professional liability, which led to a further exclusion of 50 documents not referable to medical negligence issues, but rather concerning, more specifically, veterinary and car accident liability.

The second step involved the analysis of 1117 documents exclusively relating to medical malpractice cases. For the analysis, a work grid was used to process the data, using the EXCEL program (Office 365) to systematize the data mining.

The grid was also set up with some locked fields, to minimize the inter-individual variability between the three auditors. The items present in the columns of the excel grid were: judgment no., occurrence year, publication year of the judgment, medical specialty involved, type of negligence/liability sued and recognized, type of damage (injury/death), type of parties involved (public/private facility or single healthcare worker), outcome of the trial, and compensation paid. At the end of this step, 156 judgments concerning informed consent omission or violation were investigated (Figure 1).

## 3. Results

In the 5-year period in question, 156 lawsuits were found in which compensation was claimed, as the alternative, for the infringement (i.e., omission or poor consent) by health professionals of the obligation to provide comprehensive information to patients, which is 14% of the total of judgements in the medical professional liability field, most of which were introduced in 2013–2014 (Figure 2). The average duration of the legal proceedings (period between the date of registration of the lawsuit and the date of issue of the judgment) was equal to 4.6 years.

Liability from omitted or poor consent with related compensation for damages was explicitly awarded in 24/156 judgments, or 15.4% of the total claims. There was a significantly lower conviction rate compared with the total medical malpractice convictions of the period in question, which amounted to 55% [16]. With regard to the total of the analysed judgments, the convictions for omitted or poor consent were 2.14%.

When considering the medical specialties most frequently involved in medical liability for breaching the informed consent regulation, the leading role of those related to surgery was noted. In particular, general surgery was involved in 31 cases, plastic surgery and orthopaedics in 20 cases each, gynaecology-obstetrics in 17, and neurosurgery and dentistry in 12 cases each. The conviction rate for the above-mentioned specialties varied from a minimum of 15% (orthopaedics) to a maximum of 33.3% (neurosurgery) (Figure 3).

With reference to the informed consent obligation breaching, it should be noted that although in 22/156 cases the Judge and the Technical Consultant found some kind of deficiency at the time of obtaining informed consent, no damage was eligible for compensation.

As regards the type of infringement and considering all the legal cases in which it was detected (also including those in which no damage was found to be eligible for compensation), in 10 cases we observed the absence of any evidence concerning informed consent, while in 41 cases, the informed consent forms attached to the medical record were considered vague, inconsistent and improper for providing the information needed (Figure 4). In such cases, the forms were usually generic, pre-printed and, in two cases, lacking in the doctor’s and/or patient’s signature.

Finally, the data relating to the amount of compensation paid for damages from the infringement both of informed consent obligations (Table 1) and the patient’s right to self-determination were extracted. The total amount paid over the years of investigation was EUR 287,144.59, not including interest for late payment, legal fees and the judge’s consultant’s fee.

## 4. Discussion

As noted in the introduction, informed consent has now taken on the characteristics of a legal act, which, in order to be relevant and valid, must necessarily be provided by individuals with this right (i.e., medical decision-making capacity and being free to act), entitled to provide consent and to appreciate the meaning of the related implications, including benefits and risks, of a specific medical procedure recommended to them. In the health field, therefore, respect for the patient’s decision-making autonomy is made manifest by the process underlying informed consent.

According to the legal medical doctrine, informed consent must be personal, free, current, expressed, aware, required, specific, participatory and revocable at any time. In this regard, some authors [17] raised concern about informed consent forms delivered to patients: they cannot always be considered effective tools for an informed choice or for a discharge from medical liability.

In this context, criticism is aimed at the information session which is no longer or not so much a time spent with the patient for disclosing information about the medical treatment path, but rather a moment for the fulfilment of legal and bureaucratic formalities [18]. In some circumstances, this undue attention to the legal aspects has led to an increasing loss of the real meaning of informed consent, which is the certainty that the patient has really understood what is proposed to him/her in terms of a diagnostic-therapeutic path.

It should also be noted that this deviation, in which the information session and conversation with the patient takes second place compared to the mere signing of forms, has a dangerous impact, as a boomerang effect, on the quality of the medical care and, under some circumstances, even on the outcomes of the medical treatments, thus paradoxically increasing the potential risk of medical malpractice lawsuits.

Some authors have pointed out the so-called “anticipatory anxiety” phenomenon, to emphasize the impact (e.g., in the surgical field) that complete and comprehensive information during the pre and post operative care period can have on the ability to manage and deal with surgical anxiety. According to this approach, a patient who is properly informed on what to expect would develop a sort of “psychological immunity” [18] with evident positive effects also, for example, on the postoperative pain reduction and the consequent need to take pain-relievers, together with a more rapid discharge [19] and, therefore, a higher level of general satisfaction with the health service received.

If on the one hand what was said above shows the importance and usefulness of informing patients properly and comprehensively, on the other, against these results, some authors highlight how the disclosure of clinical information relating, for example, to surgery, might have negative effects known as the “nocebo effect” even when they are completely non-attributable to any doctor’s liability.

The nocebo effect is a phenomenon occurring when patients, being aware of possible side effects in response to a medical treatment, experience a possible worsening of their clinical conditions [20]. It is believed that the nocebo effect can be triggered by the patient’s anticipatory anxiety and his/her expectation of adverse effects [21].

From an ethical perspective, the onset of such an effect could represent a dilemma between the obligation to inform and the duty to protect patients’ health avoiding adverse effects [21].

The results of this research have shown that liability for omissions in informed consent provision, even though outnumbered if compared to other medical malpractice lawsuits, cannot be underestimated in light of the indirect impact that omitted or lacking information can determine in terms of dissatisfaction and quality of care. Furthermore, it is crucial, in order to limit the spread of the phenomenon and to reduce, if possible, the impact of medical malpractice, to carry out an in-depth analysis aiming at highlighting the causative roots of the issue.

Regarding the violation of informed consent-related medical malpractice, in the general climate of conceptual and jurisprudential evolution delineated in the introduction, the Italian Court of Cassation in 2019 has detected several hypotheses that may result from omitted or poor information as indicated briefly below: “

*a)* 
*Omitted/poor information attributable to any medical procedure to which the patient would not have undergone, in the same health conditions, hic et nunc but that harmed the patient’s health resulting from the physician misconduct. In this case, the compensation shall be limited only to direct health damages suffered by the patient, in its double aspect, moral and relational;*
*b)* 
*Omitted/poor information attributable to any medical procedure that harmed the patient’s health resulting from the physician misconduct and to which the patient would not have undergone. In this case, the compensation shall concern the patient’s right to health and self-determination;*
*c)* 
*Omitted/poor information attributable to any medical procedure that harmed the patient’s health (even in case of a worsening of his/her pre-existing health conditions) due to the non-negligent conduct of the physician and to which the patient would not have undergone. The compensation shall be paid on an equitable basis according to the breach the right to self-determination regulation. On the contrary, the health damage (to be intended in any case in causation to the medical misconduct, since in the event of a proper information the medical procedure would not have been performed) shall be judged with respect to any possible “differential” situation …*
*d)* 
*Omitted information related to a medical procedure that did not harm the patient’s health, to which he/she would not have undergone anyway. In this case, no compensation shall be paid;*
*e)* 
*Omitted diagnosis/misdiagnosis that did not harm the patient’s health, but which nevertheless prevented him/her from accessing more accurate and reliable medical examinations. In this case, the damage from the breach of the right to self-determination shall be unrefunded unless the patient’s allegations show that he/she suffered from harmful consequences, of a non-pecuniary nature, in terms of subjective suffering following a worsening of physical and psychological health due to the omitted, improper or poor information… “(Italian Court of Cassation decision no.28985/2019).*


With this decision, the Supreme Court (which is at the apex of ordinary jurisdiction in Italy) has in practice outlined a jurisprudential guideline to be followed in the field of professional liability deriving from a lack of or inadequate consent.

What emerged in most of the analysed lawsuits was the vagueness of the information forms provided for patients’ attention, especially the paucity of information regarding any possible alternative treatment option (considered as information to be provided as per Law 219/2017) with respect to the one proposed and the absence of data on risks and complications, when reasonably predictable.

It should be noted that in the literature the legal doctrine regarding informed consent tends to focus excessively on mere allegations proving the provision of information without taking into account patients’ comprehension level. In this regard, it has been reported that the failure rate in terms of understanding by patients exceeds 90% [22]. This suggests one of the reasons that may underlie the claim, also with reference to the cases we have analysed. Indeed, it is possible that some of the cases for which lawsuits are proposed for breaches of consent may simply be induced by a low level of understanding on the part of the patient, a situation that health care providers should take into account as potentially avoidable.

However, it should be remembered that medical science is not always able to obtain adequate levels of certainty such as to ensure a clear and precise decision-making process. In the literature, several authors have dealt with the importance of being able to communicate even uncertainty about the clinical outcomes, especially when it is difficult, on the basis of scientific data, to ensure a complete healing.

On such occasions, also for the preventive purposes of any future medical malpractice claims, it is advisable to achieve a real therapeutic alliance, based not on the mere attachment of forms which are sometimes of poor medico-legal value, but rather on a relationship recognizing and sharing the cognitive and affective implications of uncertainty and in which informed consent may have a key clinical role [23].

The data obtained concerning the medical specialties mostly involved in medical malpractice claims, due to the breach of the informed consent regulation, confirmed what was reported by other authors on the prevalence of those associated with surgical procedures [24]. In fact, in the surgical context, the medical act is characterized by a certain degree of invasiveness and risk of failure and in which the achievement of an adequate awareness on the part of the patient is deeply required, especially with regard to the realistically obtainable outcomes.

The examination of the convictions rate with respect to the total number of legal cases, has shown interesting data relating to neurosurgery: although it is the third medical specialty by frequency and number of lawsuits, it presents a higher risk of being charged with damages resulting from the lack of informed consent. According to a US study, liability for damages from the infringement of informed consent regulation in neurosurgery for brain tumour represents the second most common reason for medical malpractice claims, with the lack of diagnosis being the unique frequent cause [25].

With regard to the compensation paid, the data reported, if compared with previous Italian research carried out on a similar case study [24], show an increase in the amount of compensation paid for this specific type of damage as well as in the average amount paid. The criterion used was purely equitable, i.e., based on the so-called “equitable discretion of the court”, a parameter subject to criticism as a result of a certain degree of unpredictability of the payable compensation as shown by the results of the present case study in which the compensation paid ranges between EUR 2000 and 50,000.

This rise, represented both by the number of sentences and by the entity of the relative compensation paid, could be explained by a greater awareness of the importance of this aspect in the clinical and, above all, legal field (also evidenced by the introduction of a dedicated law in the Italian legal system) and by the general change in the figure of the patient-citizen, a subject who appears to be the depositary of ever greater rights, first and foremost, in the health setting, the right to health and self-determination. A further interesting aspect emerging from the judgments is related to the exclusion of medical procedures from certain insurance policies in the event of damages following the lack of informed consent, with healthcare professionals taking on the duty to compensate the damage out of their own pockets. This kind of exclusion from healthcare policy might change the final goal to obtain informed consent, reducing it to a moment of mere legal significance, useful for avoiding lawsuits for professional liability.

## 5. Conclusions

The duplicity of the concept of informed consent has now become a reality that we should all be aware of. In fact, if on the one hand, from a strictly clinical point of view, it represents a moment of conversation and sharing between the physician and the patient, the real foundation of a conscious relationship of care, on the other, it should be considered that the informed consent doctrine was born and consolidated in the legal and jurisprudential field.

The research carried out has highlighted how any breach or lack of informed consent may lead to legal proceedings and the related implications in terms of compensation. The topic, rarely addressed in the literature, appears to be extremely up to date, as healthcare systems are facing a real social revolution, in which patients’ demands for health also involve a greater emphasis on patient involvement and the consequent respect for their decision-making autonomy.

It should be emphasized that the prejudice resulting from the violation of the patient’s right to self-determination due to a lack of or inadequate information by the health care provider, represents a form of professional liability that is entirely foreseeable. Just as preventable are the related costs, represented by both the compensation paid and the associated expenses (court costs), which in most cases burden the budgets of healthcare facilities (in Italy, mainly public and therefore state-funded) and individual healthcare professionals.

The data relating to the reasons given in the judgment were particularly significant. In fact, it has been pointed out that one of the most critical issues is represented by the vagueness and generic nature of the information form provided. Therefore, it seems clear that informed consent should not be considered as a mere signature on a sheet of paper. It is precisely the preventability of such cases that makes it obvious for healthcare facilities and professionals, in order to avoid unnecessary compensation, to equip themselves with all those tools, including communication tools, which make it possible to guarantee the construction of a solid therapeutic alliance with the patient.

## Figures and Tables

**Figure 1 ijerph-20-02118-f001:**
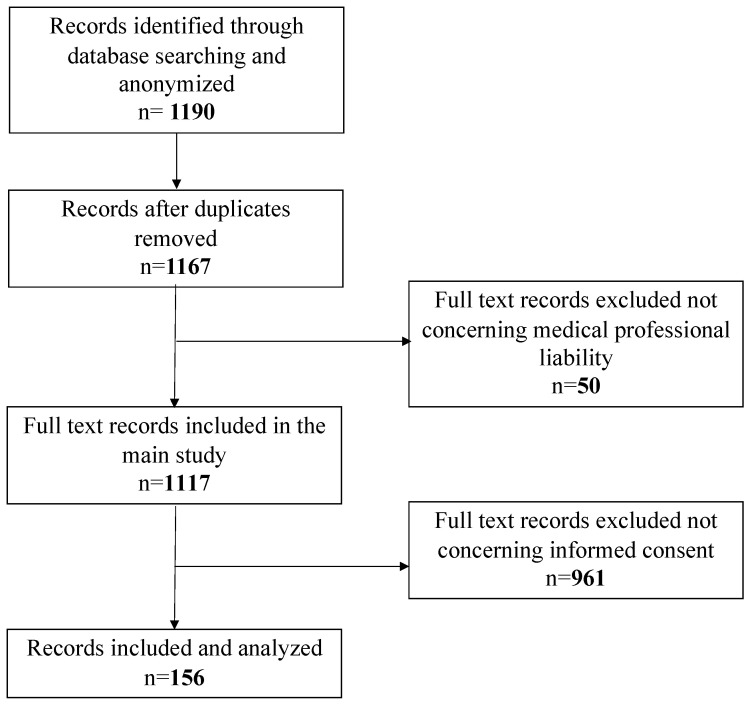
Flowchart reassuming the process of document selection.

**Figure 2 ijerph-20-02118-f002:**
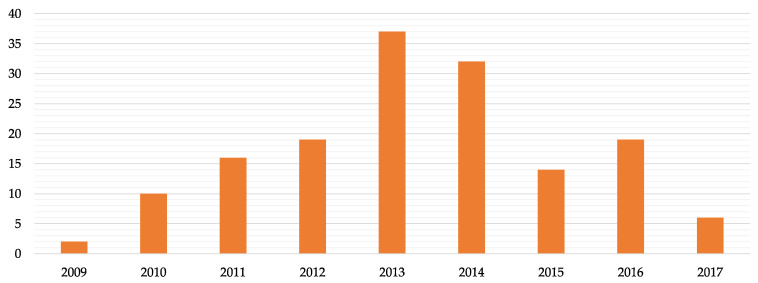
Number of judgements introduced per year.

**Figure 3 ijerph-20-02118-f003:**
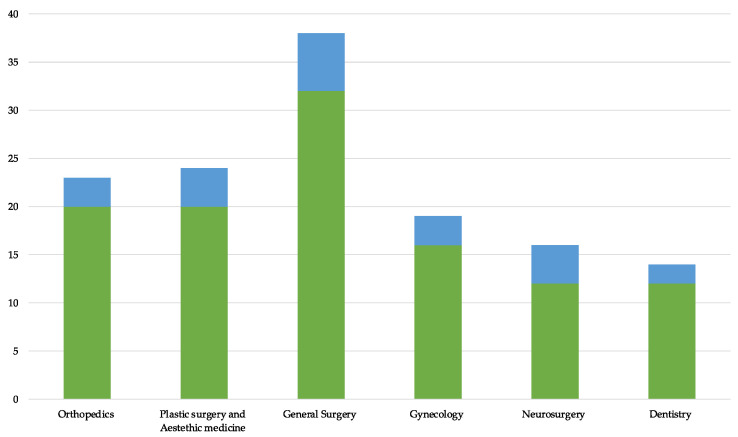
Number of causes (green) divided per branch and related convictions (blue).

**Figure 4 ijerph-20-02118-f004:**
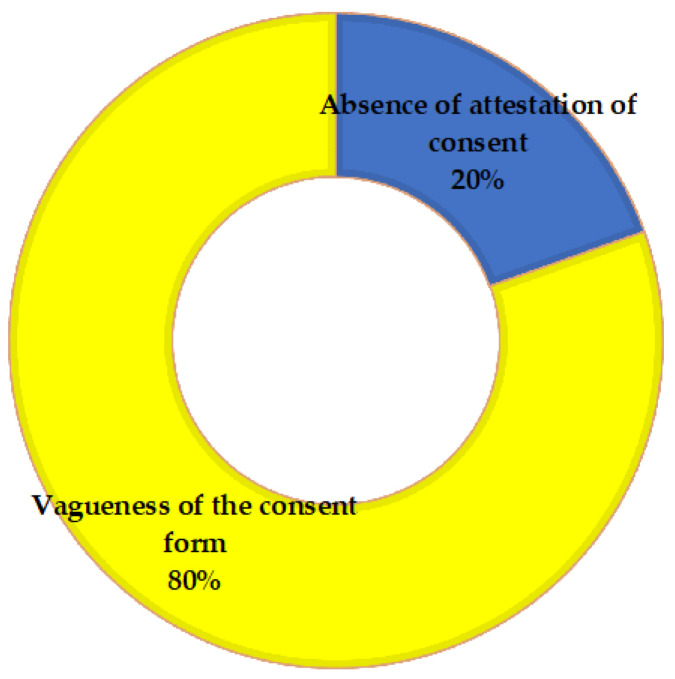
Cases of violations of consent admitted by the Court.

**Table 1 ijerph-20-02118-t001:** Compensation paid in the period of investigation.

Total Amount Paid (2016–2020)	Mean	Minimum	Maximum
**EUR 287,144.59**	EUR 11,964.35	EUR 2,000	EUR 50,000

## Data Availability

Not applicable.

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
