# Peer review of "Informed Consent: Legal Obligation or Cornerstone of the Care Relationship?"

_ijerph, 2023, doi:10.3390/ijerph20032118_

Round 1

Reviewer 1 Report

This article treats an actual problem of the violation of the patient's right to self-determination due to lack of or inadequate information by the health care provider. The authors specifically focused on the question of the informed consent regulations through the analysis of the judgments from the Civil Court of Rome. This paper seeks demonstrate the importance and usefulness of informing patients properly and comprehensively. The research findings are significant for the healthcare professionals in Italy.

After reading the manuscript and paying particular attention to the data and methods, I rated the manuscript on attributes using the MDPI scorecard. The eventual recommendation of mine is to accept the paper after minor revision.

The title of a research paper should be interesting to the reader. It begins with a catchy main title and is followed by a subtitle that gives information about the subject of the manuscript. That’s fine. A good title is memorable. In order to keep the title statement as concise as possible I would like to recommend deleting the explanation “Analysis of legal cases from an Italian court”.

The abstract contains information to enable the reader to understand what was done. My suggestions: (1) to delete unnecessary words in abstract (Introduction, Methods, Results, Conclusions); (2) to demonstrate clearly the importance of the paper for the field.

The keywords accurately reflect the content.

The Introduction provides the reader with a conceptual origins of the informed consent and articulates aim of the article. The Introduction establishes the originality of the article by demonstrating the need for investigations in the healthcare field on the informed consent as a legal act. According to me, Introduction should shortly summarize recent research related to this topic, too.

The data collection and methods are detailed enough. The authors give explicit description of how data were analysed.

The text of Results is clear and easy to read.

The Discussion focuses on explaining the importance and usefulness of informing patients properly and comprehensively. The authors effectively demonstrate their ability to think critically about an issue of omissions in informed consent provision.

The publication would make a useful contribution to the improvement of healthcare policy. However, the concluding part of the article could be communicated better. The conclusions should be based on the evidence and arguments presented. The key messages should be evidence-based, i.e. they should clearly describe what the data show.

Reviewer 2 Report

The information was clearly presented in this retrospective case analysis of suits related to questionable informed consent cases.

It is important informative most especially the identification of highly litigious fields. The graph nicely portrays this.

Reviewer 3 Report

I read the proposed manuscript with great interest. The article is well-written and fits into a very relevant research area. 

However, there are a few aspects that the authors should clarify. 

In general I have some suggestions about the structural assessment of the text. In Introduction, you have to display the prospective

Firstly, in the discussion you must improve your analysis based on results. In other words you must describe the field in which you conduct the study. An historiographical connection can be useful but ethical description is not enough to build a scientific paper.  First of all, you have to move the legal basement of lines 204-215 before the "Italian Court of Cassation decision no.28985 / 2019). ". Line 76 -79: is this a cit? In this case, put quotes. 

I suggest you to re-write introduction following this schedule: 

1. Historiographical description and definitions. 

2. Italian context and law about informed consent [Constitution -> Constitutional Court Sentences -> Italian Law 219/17 ]. Here, some citations are required to improve you bibliography: 

D'Imperio, A et al. “Uninformed consent: Who knows what Ivan Ilyich would have thought?.” La Clinica terapeutica vol. 172,4 (2021): 264-267. doi:10.7417/CT.2021.2328

Viola, R V et al. “Rules on informed consent and advance directives at the end-of-life: the new Italian law.” La Clinica terapeutica vol. 171,2 (2020): e94-e96. doi:10.7417/CT.2020.2195

Turillazzi, E et al. “Physician-Patient Relationship, Assisted Suicide and the Italian Constitutional Court.” Journal of bioethical inquiry vol. 18,4 (2021): 671-681. doi:10.1007/s11673-021-10136-w

3. The "san martino" decision is mandatory, BUT you may cite only this sentence because you study concern a period of 2016 - 2020 and the decision was of 2019!

4. Consequences of the law on informed consent to treatment as the aim of the study. 

I congrats whit you for your great and useful work on your results. 

About discussion I have many others suggestions. 

In this paragraph you have to discuss your data, derived from your results. You need to analyze your data from the context explained in the introduction. In other words, you have to answer the question: why there are so many compensations? What are the foundations on which compensation is based? Where does the compensation burden come from?Here you can insert the San Martino sentence, with an improvement of your bibliography about the obvious conseguences: 

Fineschi, Vittorio et al. “Defensive Medicine in the Management of Cesarean Delivery: A Survey among Italian Physicians.” Healthcare (Basel, Switzerland) vol. 9,9 1097. 25 Aug. 2021, doi:10.3390/healthcare9091097

Happy to look at a revised version.

Round 2

Reviewer 3 Report

Thank you for submitting the revised version. 

Authors improved the manuscript following the indications. Could be accepted. 

Congratulation.